# High Sensitivity C Reactive Protein in Patients with Rheumatoid Arthritis Treated with Antibodies against IL-6 or Jak Inhibitors: A Clinical and Ultrasonographic Study

**DOI:** 10.3390/diagnostics12010182

**Published:** 2022-01-13

**Authors:** Beatriz Frade-Sosa, Andrés Ponce, Virginia Ruiz-Esquide, Maria Jesús García-Yébenes, Rosa Morlá, Nuria Sapena, Julio Ramirez, Ana Belen Azuaga, Juan Camilo Sarmiento, Juan D. Cañete, Jose A. Gomez-Puerta, Raimon Sanmarti

**Affiliations:** 1Department of Rheumatology, Hospital Clinic of Barcelona, 08036 Barcelona, Spain; frade@clinic.cat (B.F.-S.); aponce@clinic.cat (A.P.); vruizesq@clinic.cat (V.R.-E.); morla@clinic.cat (R.M.); sapena@clinic.cat (N.S.); fjramirez@clinic.cat (J.R.); abazuaga@clinic.cat (A.B.A.); sarmiento@clinic.cat (J.C.S.); jcanete@clinic.cat (J.D.C.); jagomez@clinic.cat (J.A.G.-P.); 2Instituto de Salud Musculoesquelética, 28045 Madrid, Spain; mjgarciadeyebenes@inmusc.eu

**Keywords:** IL-6 inhibitor, JAK inhibitor, rheumatoid arthritis, ultrasound synovitis, high-sensitivity C-reactive protein

## Abstract

Background: We examined whether high-sensitivity CRP (hsCRP) reflected the inflammatory disease status evaluated by clinical and ultrasound (US) parameters in RA patients receiving IL-6 receptor antibodies (anti-IL-6R) or JAK inhibitors (JAKi). Methods: We conducted a cross-sectional study of patients with established RA receiving anti-IL-6R (tocilizumab, sarilumab) or JAKi (tofacitinib, baricitinib). Serum hsCRP and US synovitis in both hands were measured. Associations between hsCRP and clinical inflammatory activity were evaluated using composite activity indices. The association between hsCRP and US synovitis was analyzed. Results: 63 (92% female) patients (42 anti- IL-6R and 21 JAKi) were included, and the median disease duration was 14.4 (0.2–37.5) years. Most patients were in remission or had low levels of disease. Overall hsCRP values were very low, and significantly lower in anti-IL-6R patients (median 0.04 mg/dL vs. 0.16 mg/dL). Anti-IL-6R (82.4%) patients and 48% of JAKi patients had very low hsCRP levels (≤0.1 mg/dL) (*p* = 0.002). In the anti-IL-6R group, hsCRP did not correlate with the composite activity index or US synovitis. In the JAKi group, hsCRP moderately correlated with US parameters (r = 0.5) but not clinical disease activity, and hsCRP levels were higher in patients with US synovitis (0.02 vs. 0.42 mg/dL) (*p* = 0.001). Conclusion: In anti-IL-6R RA-treated patients, hsCRP does not reflect the inflammatory disease state, but in those treated with JAKi, hsCRP was associated with US synovitis.

## 1. Introduction

Rheumatoid arthritis (RA) is a chronic progressive disease characterized by inflammation of the synovial joints, leading to joint destruction and disability that can be prevented by promptly initiated and effective therapy [1]. To ensure therapy is effective, regular clinical evaluations are needed. C-reactive protein (CRP) is an acute-phase reactant (APR) synthesized by hepatocytes on stimulation by pro-inflammatory signals, including cytokines such as IL-6. CRP serum levels reflect inflammatory activity in patients with RA and other forms of arthritis and are used to monitor inflammatory activity and the response to therapy in RA, and form part of composite activity indices (e.g., disease activity score (DAS), simplified disease activity index (SDAI), etc.) used to evaluate disease activity in clinical trials and practice [2,3]. Furthermore, persistent CRP elevations in RA patients are associated with progressive joint damage [4,5].

Monoclonal antibodies against IL-6 receptors (anti-IL-6R) (tocilizumab and sarilumab) and JAK inhibitors (JAKi) (tofacitinib, baricitinib, and, more recently, upadacitinib) are included in the treatment strategy of patients who do not achieve the therapeutic goal (remission or low disease activity) with conventional synthetic diease modifying antirheumatic drugs (DMARDs) (e.g., methotrexate) [6]. In patients treated with anti-IL-6R, CRP does not satisfactorily reflect the degree of inflammation, since their production is aborted with these biological therapies, without this being reflected in a substantial improvement in synovitis [7]. Although less well known and studied, this also occurs in patients treated with JAKi, which also partially inhibits IL6 signaling [8]. Therefore, it has been questioned whether CRP reflects the inflammatory state in patients receiving these therapies. 

Joint ultrasound (US) is a validated imaging technique for synovitis evaluation in RA, with a higher sensitivity than the clinical examination [9]. In recent years, US has revealed that a significant percentage of patients classified as being in clinical remission may exhibit active synovitis in US [10,11]. 

Our objective was to analyze whether high-sensitivity CRP (hsCRP) reflects the inflammatory state of RA evaluated by clinical and US parameters in RA patients in real clinical practice receiving anti-IL-6R and JAKi. We hypothesized that there is no association in patients treated with anti-IL-6R due to the dramatic impact on hsCRP serum levels with these therapies whereas, in patients treated with JAKi, hsCRP may be a sensitive biomarker of disease activity. 

## 2. Materials and Methods

### 2.1. Design and Study Population

This observational cross-sectional study included consecutive RA patients (ACR/EULAR 2010 criteria) [12] from our Arthritis unit receiving anti-IL-6 receptor mAb (tocilizumab or sarilumab) or JAKi (baricitinib or tofacitinib) for >3 months. Demographic data, disease duration, autoantibodies (ACPA and/or RF), radiographic erosive disease, previous biologic drugs, and concomitant therapy were collected. Patients were excluded if they presented signs of active infection or another clinical condition that, in the opinion of the investigator, could alter the hsCRP result. 

### 2.2. Measurement of Clinical Disease Activity and Assessment of Blood Biomarkers

Before US assessment, all patients underwent clinical assessment, including 28 swollen and tender joint counts (28SJC and 28TJC) and physician and patient global assessment (PhGA and PGA) with visual analog scales (0–10). Three composite disease activity indices were subsequently calculated: DAS28, SDAI, and clinical disease activity index (CDAI). Patients were asked to complete two questionnaires: the Health Assessment Questionnaire (HAQ) and the Routine Assessment of Patient Index Data 3 (RAPID3). 

Blood samples were obtained at the clinical evaluation. hsCRP was determined using an immunoturbidimetric method measured using Siemens Atellica^®^ Solution (lowest detection limit of 0.02 mg/dL: Normal value (NV) < 0.4 mg/dL). hsCRP serum levels < 0.1 mg/dL were considered very low.

### 2.3. Imaging Biomarkers: Ultrasound Score

An experienced sonographer (AP), who was unaware of the clinical joint examination, performed the US evaluation. High sensitivity US equipment was used for US evaluations (MyLab9^®^; Esaote, Genoa, Italy) with a frequency range of 12–14 MHz and a pulse repetition frequency between 900 and 1000 Hz. Definitions of Outcome Measures in Rheumatoid Arthritis Clinical Trials (OMERACT) were used to describe US findings [13]. Synovial hypertrophy (SH) and intra-articular power Doppler (PD) signaling were evaluated according to EULAR guidelines [14] and the SH and PD signals were graded using a four-grade semi-quantitative scoring system (0  =  no, 1  =  mild, 2  =  moderate, and 3  =  severe) according to the methodology of Szkudlarek et al. [15]. Eleven joints and tendons of each hand (including the proximal interphalangeal joints, metacarpophalangeal joints, and wrists) were assessed, and the highest SH and PD grade detected during the scans was adopted as representative of each joint, respectively.

The definition of active synovitis was SH grade ≥ 2 plus PD signal ≥ 1 in at least one joint. We calculated an SH score (sum of SH scores in all joints, range 0–66), a PD score (sum of PD scores in all joints, range 0–66), and a global US score (sum of the PD and SH scores, range 0–132) for each patient [16].

### 2.4. Statistical Analysis

Continuous data were presented as median (range) and categorical variables as absolute frequency with percentages. Groups were compared using parametric or nonparametric tests according to the distribution of the variables. Spearman’s correlation coefficient was used to assess the association between hsCRP and clinical disease activity and US scores (PD score, SH score, and global score). The analysis was made using STATA version 12 (STATA Corp, College Station, TX, USA). 

The study was conducted in accordance with the Declaration of Helsinki and was approved by the Clinical Research Ethics Committee of the Hospital Clinic of Barcelona (Reg. HCB20210783). Written informed consent was obtained from all patients before study enrolment.

## 3. Results

### 3.1. Demographic, Clinical, and Therapeutic Characteristics 

Sixty-three patients were included (42 receiving anti-IL-6R and 21 JAKi): 92% were female with a median age of 58.6 (26.4–84.7) years and a median disease duration of 14.4 (0.2–37.5) years. Ninety percent were seropositive (RF and/or ACPA), and 75.8% had erosive disease (Table 1). In general, disease activity was low, with a median CDAI value of 10 (0–41). The median duration of drug therapy was 27.9 (2.9–139.9) months, with significant differences in patients receiving anti-IL-6R or JAKi (median 43.8 months vs. 9.9; *p* < 0.001) (Table 2).

hsCRP serum levels were low in both groups, although significantly lower in patients receiving anti-IL-6R (0.04 mg/dL vs. 0.16 mg/dL; *p* < 0.001). The percentage of patients with very low hsCRP levels (values ≤ 0.1 mg/dL) was 81% in the anti-IL-6R group and 42.9% in the JAKi group. Significant between-group differences in ESR and Hb levels were also observed (Table 2).

No significant between-group differences were observed in clinical disease activity (28TJC, 28SJC, CDAI, SDAI, HAQ, and RAPID3). The DAS28 was higher in the JAKi group (2.35 vs. 3.44; *p* < 0.001). No significant between-group differences were observed in US scores (Table 2).

### 3.2. Correlation between hsCRP and Disease Activity

We studied the correlation between hsCRP and the clinical and patient reported outcomes (PRO) and US scores. No correlations between hsCRP and any clinical activity index or PRO were observed in the anti-IL-6R group (r < 0.2). In the JAKi group, there was a trend to a positive correlation (r > 0.35) with some parameters of clinical activity, such as 28SJC, although they were not significant. With the US parameters, a significant positive correlation was found only in the JAKi group, even though this was moderate (r = 0.5) (Table 3).

### 3.3. hsCRP Serum Levels in Patients with and without Ultrasound Synovitis

Active US synovitis (SH grade ≥ 2 plus PD signal ≥ 1) was observed in 46 patients (73%): 30 patients (71%) and 16 patients (76%) in the anti-IL-6R and JAKi groups, respectively. Classification of patients according to the presence of active US synovitis showed a trend to higher clinical disease activity in patients with US synovitis, especially PhGA and 28SJC (Table 4). No differences in hsCRP levels were observed in patients with and without US synovitis in the anti-IL-6R group. However, JAKi patients with active US synovitis had a higher hsCRP than those without (Table 4, Figure 1).

Classification of patients according to CDAI activity (remission/low activity (≤10) vs. moderate/high activity (>10)), showed no significant differences in hsCRP values in either group (Table 5). 

## 4. Discussion

We evaluated the association between hsCRP and clinical and US disease activity in RA patients receiving anti-rIL6 and JAKi. In anti-rIL6 patients, hsCRP did not reflect the presence and amount of synovitis, whereas, in patients receiving JAKi, hsCRP showed a moderate correlation with US synovitis but not with clinical disease activity. 

CRP serum concentrations are included in composite indices of clinical disease activity in RA, such as DAS28CRP or SDAI, which are used in clinical trials and daily clinical practice. Significant reductions in CRP and composite joint scores using this APR have been observed in patients treated with antirheumatic drugs in RA, including csDMARDS and targeted therapies [2]. However, biological drugs that significantly inhibit IL-6 production, such as tocilizumab or sarilumab, have a dramatic impact on the hepatic synthesis of CRP, with an impressive and rapid reduction in CRP serum concentrations that persist over time. This dramatic improvement in CRP levels does not always reflect a parallel improvement in disease activity and swollen joint counts [7]. The disproportional reduction in CRP levels in comparison with clinical measures of inflammation is a well-known phenomenon in RA patients treated with anti-IL-6R and, it has been suggested, in those treated with JAKi, due to the intracellular effects of JAKi on the IL-6 pathway. In a recent study comparing the effect on CRP reduction between tocilizumab and baricitinib, there were no significant differences between the two drugs although, numerically, the largest CRP reduction was observed with tocilizumab [8]. In another study, disproportionate rates of remission using DAS28CRP were observed in patients treated with tofacitinib, suggesting a major role for this drug in the reduction of CRP synthesis [17]. Therefore, it has been suggested that CRP serum levels and, by extension, disease activity scores containing this APR are not sensitive markers of inflammation (synovitis) in patients treated with anti-IL-6R or JAKi, in contrast to other biological therapies, such as TNF inhibitors.

We have addressed in this study the association between hsCRP with clinical disease activity but also with ultrasound synovitis in RA patients. It is well established that CRP serum levels are a good biomarker of ultrasound disease activity (positive PD signal) in RA patients treated with csDMARDs [18]. On the other hand, tocilizumab has demonstrated a profound effect on the improvement of US synovitis in patients with RA [19] as it occurs with JAKi, such as baricitinib [20]. However, no studies previously addressed the exact role of serum CRP in assessing disease activity, including ultrasonographic parameters in patients treated with tocilizumab or sarilumab, but not in patients under JAKi.

We confirmed that CRP serum levels are not a biomarker of active synovitis in patients under anti-IL-6R therapy, as shown by previous reports [21,22,23]. However, high CRP serum levels at the initiation of tocilizumab therapy have been considered a biomarker of a good clinical response [24]. The measurement of this APR as a surrogate marker of inflammation for monitoring disease activity is not recommended. Investigations by our group in RA patients receiving tocilizumab showed that CRP levels are not associated with disease activity, and CRP suppression mainly reflects detectable drug serum levels. This absolute lack of correlation between CRP and inflammatory activity was also found when synovitis was measured by US in the present study. In contrast, other serum proteins, such as calprotectin [25], a myeloid-related protein, or leucine-rich α2 -glycoprotein (LRG), have been found to be good biomarkers of inflammation in these patients [26]. 

In patients treated with JAKi, we found similar results, although some differences emerged. First, the reduction in hsCRP levels was not as dramatic as that observed with anti-IL-6R: the percentage of very low levels of hsCRP (≤0.1 mg/L) was significantly higher in patients treated with anti-IL-6R than in those with JAKi (82% vs. 48%), reflecting a less pronounced inhibition of CRP synthesis with JAKi. Secondly, although there was no association between composite activity indices and hsCRP, as occurs with anti-IL-6R therapy, a nonsignificant trend to association with the swollen joint count was observed. Most RA patients included in this study had low disease activity or were in remission, with a median swollen joint count of 1, which may underestimate the correlation between clinical synovitis and hsCRP. Thirdly, a positive correlation between US scores and hsCRP was observed in JAKi patients in contrast to those observed with anti-IL-6R, confirming that CRP may be a sensitive marker of US synovitis in these patients, a finding that may have implications in clinical practice in patients treated with these targeted therapies in apparent clinical remission or with low disease activity.

Our study has some limitations. We included a relatively small sample size, especially of patients treated with JAKi. On the other hand, the sample is representative of patients with established disease, with a relatively long-term duration of drug therapy, and with low disease activity. The accuracy of hsCRP levels as a biomarker of disease activity, measured by clinical parameters, or US in patients with JAki in other populations (early RA or with different degrees of inflammatory disease activity), should be elucidated. Other factors, such as serum drug levels or pharmacogenomics that may affect the clinical response to targeted therapies, were not examined in this study [27,28]. 

## 5. Conclusions

We confirmed that hsCRP is not a biomarker of clinical disease activity in patients with RA treated with anti-IL-6R, and does not reflect active synovitis detected by US in these patients. hsCRP levels were low in patients receiving JAKi but not as low as those observed with anti-IL-6R. Furthermore, in patients treated with JAKi, hsCRP may be a surrogate marker of synovial inflammation measured by US. Therefore, the lack of normalization of hsCRP serum levels in patients receiving JAKi may alert clinicians to the presence of persistent synovitis.

## Figures and Tables

**Figure 1 diagnostics-12-00182-f001:**
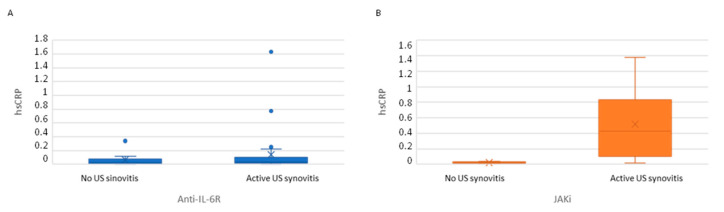
hsCRP serum levels in patients classified according to the presence or absence of active ultrasound. (**A**) Patients receiving anti-IL-6R. (**B**) Patients receiving JAKi. Active US synovitis: SH grade ≥ 2 plus PD signal ≥ 1.

**Table 1 diagnostics-12-00182-t001:** Demographic characteristics of patients by therapeutic group.

	Anti-IL-6R (*n* = 42)	JAKi (*n* = 21)	*p*-Value
DEMOGRAPHIC VARIABLES			
Age	59.9 (34.0–79.4)	52.4 (26.4–84.7)	0.145
Female, *n* (%)	39 (92.9)	19 (90.5)	1
CCP, *n* (%)	34 (91.9)	19 (95.0)	1
FR, *n* (%)	31 (83.8)	17 (85.0)	1
Erosive disease, *n* (%)	34 (81.0)	13 (61.9)	0.102
Disease duration (years)	15.7 (3.47–37.5)	12.09 (0.2–34.3)	0.040
Previous biologic treatments	1 (0–4)	1 (0–7)	0.89
Patients with previous biologic treatment, *n* (%)	25 (58.5)	11 (52.4)	0.65
CONCOMITANT TREATMENT			
Treatment duration (months)	43.8 (7.9–139.9)	9.9 (2.9–77.7)	<0.001
prednisone, *n* (%)	17 (40.5)	12 (57.1)	0.211
Prednisone equivalent dose (mg/day)	2.5 (1.25–10)	5 (1.25–10)	0.586
NSAID treatment, *n* (%)	13 (31.0)	2 (9.5)	0.60
CsDMARD treatment, *n* (%)	18 (42.9)	11 (52.4)	0.475

Anti-IL-6R: JAKi: Monoclonal antibodies against IL-6 receptors. JAK inhibitor. NSAID: Nonsteroidal anti-inflammatory drugs. csDMARD: conventional synthetic disease-modifying antirheumatic drugs. Data expressed as medians and (ranges) or total number and (percentage).

**Table 2 diagnostics-12-00182-t002:** Clinical disease activity, patient reported outcomes, and ultrasound synovitis scores by therapeutic group.

	Anti-IL-6R (*n* = 42)	JAKi (*n* = 21)	*p*-Value
28TJC	2 (0–20)	3 (0.25)	0.591
28SJC	1 (0–7)	1 (0–9)	0.580
PGA	4 (0–8.5)	4 (0–7.5)	0.321
PhGA	3 (0–7)	3 (0–7)	0.111
VAS pain	3 (0–8)	4 (0–7.5)	0.433
DAS28	2.349 (0.970–5.06)	3.439 (1.502–7.294)	0.002
CDAI	8.5 (0–31)	13 (0–41)	0.240
SDAI	8.9 (0.40–31.40)	13.40 (0.40–42.91)	0.231
HAQ	0.88 (0.0–2.88)	0.75 (0.0–2.275)	0.421
Rapid3	8.6 (0–25.50)	9.0 (1–18)	0.560
LABORATORY TESTS			
hsCRP mg/dL	0.04 (0.0–1.63)	0.16 (0.01–1.38)	0.007
ESR	5 (2–14)	16 (6–140)	<0.001
Hemoglobin g/L	143 (100–168)	125 (100–148)	<0.001
ULTRASOUND INDICES			
SH score	4 (0–18)	4 (0–28)	0.352
PD score	3.5 (0–18)	3 (0–27)	0.825
Global score	8 (0–35)	7 (0–55)	0.534

28 SJC: 28 swollen joint counts; 28 TJC tender joint count; PGA: patient global assessment, PhGA: global assessment; VAS pain: visual analog scale; DAS28: disease activity score; CDAI: clinical disease activity index; SDAI: simplified disease activity index, HAQ: health assessment questionnaire, Rapid3: routine assessment of patient index data 3. hs-CRP: high-sensitivity C-reactive protein. ESR: erythrocyte sedimentation rate. SH score: synovial hypertrophy score; PD score: power doppler score; Global score = HS + PD score. Data expressed as medians and (ranges) or total number and (percentage).

**Table 3 diagnostics-12-00182-t003:** Correlation between hsCRP and clinical and ultrasound disease activity.

	Anti-IL-6R (*n* = 42)		JAKi (*n* = 21)		Total (*n* = 63)	
	Rho	*p*-Value	Rho	*p*-Value	Rho	*p*-Value
28TJC28	−0.087	0.587	0.094	0.0684	−0.038	0.768
28SJC	−0.25	0.876	0.045	0.06	0.203	0.113
PGA	−0.191	0.876	−0.019	0.933	−0.070	0.590
PhGA	−0.171	0.298	0.376	0.093	0.117	0.369
Pain	−0.056	0.730	0.069	0.767	0.003	0.981
CDAI	−0.142	0.374	0.247	0.280	0.054	0.679
SDAI	−1.36	0.395	0.247	0.280	0.066	0.609
DAS28	−0.141	0.379	0.330	0.144	0.148	0.252
HAQ	0.12	0.938	0.270	0.236	0.065	0.613
Rapid3	−0.258	0.103	−0.005	0.984	−0.133	0.304
HS score	0.156	0.324	0.402	0.071	0.296 *	0.019
PD score	0.077	0.627	0.544 *	0.011	0.275 *	0.029
HD + PD score	0.138	0.385	0.533 *	0.013	0.296 *	0.018

28 SJC: 28 swollen joint counts; 28 TJC tender joint count; PGA: patient global assessment. PhGA: global assessment; VAS pain: visual analog scale; DAS28: disease activity score; CDAI: clinical disease activity index; SDAI: simplified disease activity index. HAQ: health assessment questionnaire. Rapid3: routine assessment of patient index data 3. hs-CRP: high-sensitivity C-reactive protein. ESR: erythrocyte sedimentation rate. SH score: synovial hypertrophy score; PD score: power doppler score; Global score = HS + PD score. Rho: Spearman correlation. * If *p* value < 0.05.

**Table 4 diagnostics-12-00182-t004:** Clinical and ultrasound disease activity according to the presence or absence of ultrasound synovitis in anti-IL-6R and Jaki groups.

	Anti-IL-6R (*n* = 42)			JAKi (*n* = 21)		
	No Active US Synovitis (*n* = 12)	Active US Synovitis (*n* = 30)	*p*-Value	No Active US Synovitis (*n* = 5)	Active US Synovitis (*n* = 16)	*p*-Value
28TJC	0.5 (0–15)	2 (0–20)	0.153	3 (0–9)	1.5 (0–25)	0.398
28SJC	0 (0–1)	1 (0–7)	<0.001	0 (0–1)	2 (0–9)	0.075
PGA	3 (0–8.5)	4 (0–6)	0.773	2.5 (2–7)	4.75 (0–7.5)	0.398
PhGA	1 (0–4)	3 (0–7)	0.017	1 (0–3)	4 (0–7)	0.011
DAS28	2.074 (0.97–4.827)	2.537 (0.970–5.064)	0.146	3.216 (1.534–4.787)	3.780 (1.502–7.293)	0.313
CDAI	6.5 (0–25)	10 (0–31)	0.052	6 (3–20)	14 (0–41)	0.313
SDAI	6.9 (0.4–25.4)	10.40 (0.40–31.40)	0.052	6.4 (3.4–20.4)	14.475 (0.40–42.91)	0.313
hsPCR mg/dL	0.03 (0–0.34)	0.04 (0–1.63)	0.417	0.02 (0.01–0.04)	0.42 (0.02–1.38)	0.001
ESR	5 (3–14)	5.50 (2–14)	0.923	14 (6–15)	20.50 (7–140)	0.062
ULTRASOUND INDICES						
SH score	0 (0–2)	6 (2–18)	<0.001	2 (0–3)	6.5 (2–28)	0.002
PD score	0 (0–1)	4 (1–18)	<0.001	0 (0–0)	5 (1–27)	<0.001
Global score	0 (0–2)	10 (3–35)	<0.001	2 (0–3)	12.5 (4–55)	<0.001

Active US synovitis: SH grade ≥ 2 plus PD signal ≥ 1. 28 SJC: 28 swollen joint counts; 28 TJC tender joint count; PGA: patient global assessment. PhGA: global assessment; VAS pain: visual analog scale; DAS28: disease activity score; CDAI: clinical disease activity index; SDAI: simplified disease activity index. hs-CRP: high-sensitivity C-reactive protein. ESR: erythrocyte sedimentation rate. SH score: synovial hypertrophy score; PD score: power doppler score; Global score= HS + PD score. Data expressed as medians and (ranges) or total number and (percentage).

**Table 5 diagnostics-12-00182-t005:** Patients classified according by group and disease activity according to CDAI.

	Anti-IL-6R (*n* = 42)			JAKi (*n* = 21)		
	CDAI ≤ 10 *n* = 25	CDAI > 10 *n* = 17	*p* Value	CDAI ≤ 10 *n* = 8	CDAI > 10 *n* = 13	*p* Value
**hs-PCRP mg/dL**	0.035 (0.0–0.77)	0.04 (0.0–1.63)	0.38	0.09 (0.01–1.08)	0.34 (0.02–1.38)	0.57
**Hs-CRP ≥ 0.1 mg/dL**	4 (16)	4 (23)	0.51	4 (50)	8 (61.5)	0.27

CDAI: clinical disease activity index, hs-CRP: high-sensitivity C-reactive protein. Data expressed as medians and (ranges) or total number and (percentage).

## Data Availability

The data presented in this study are available on request from the corresponding author.

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
