# Peer review of "High Sensitivity C Reactive Protein in Patients with Rheumatoid Arthritis Treated with Antibodies against IL-6 or Jak Inhibitors: A Clinical and Ultrasonographic Study"

_diagnostics, 2022, doi:10.3390/diagnostics12010182_

Round 1
Reviewer 1 Report
To the authors,
I think this is a well-written paper.
However, I don't know why the authors designed this research plan.
Since IL-6 is an inducer of CRP, the direct inhibition of IL-6 function can modify CRP levels.
Even with increased sensitivity, it is difficult to determine what the results mean.
When CRP is detected, there are various confounding factors such as the dosage of the drug, blood concentration of the drug, and increased IL-6 production. I don't see the point in changing the measurement sensitivity.
For reference, LRG (Leucin rich alfa2 glycoprotein) is now considered to be useful for monitoring inflammatory response during anti-IL-6R antibody therapy.
Minor points
1) The IL-6 receptor is rarely described as rIL-6. In a search using PubMed, there are 32 papers that use the term "rIL-6" within the 10 years from 2011 to 2021. With the exception of one episode in which "rIL-6" indicated rat IL-6, the rest were all used in the sense of recombinant IL-6.
IL-6 recepter is commonly referred to as IL-6R.
2) Please check the labels of each table. Some table present "ant-" and the other present "anti-". ("anti- would be correct. Again, the term "anti-rIL-6" is not standard description.)
Author Response
I think this is a well-written paper. However, I don't know why the authors designed this research plan. Since IL-6 is an inducer of CRP, the direct inhibition of IL-6 function can modify CRP levels. Even with increased sensitivity, it is difficult to determine what the results mean. When CRP is detected, there are various confounding factors such as the dosage of the drug, blood concentration of the drug, and increased IL-6 production. I don't see the point in changing the measurement sensitivity. For reference, LRG (Leucin rich alfa2 glycoprotein) is now considered to be useful for monitoring inflammatory response during anti-IL-6R antibody therapy.
We thank the reviewer for the feedback. The objective of the study was to ascertain the role of CRP in assessing disease activity (clinical and ultrasonographic) in RA patients treated with drug therapies with an important impact on CRP synthesis. Both biological therapy with anti-IL-6R therapy or targeted therapy with JAKi (but to a lesser extent) have this effect on CRP serum levels and which may not be an adequate biomarker of disease activity in these patients. We investigated using hsCRP, trying to analyze the role of low serum levels of CRP that can be measured with this more sensitive technique, and whether this APR may be useful to identify synovitis (clinical or by ultrasonography) in these patients and compare the two groups (anti-IL-6R or JAKi). Our results confirm that hsCRP is not useful as a measure of disease activity in anti-IL-6R patients but may be a biomarker of ultrasound synovitis in patients taking JAKi.
We agree with the reviewer that some confounding factors may be relevant, as described, although, in the case of anti-ILR6, the most important factor is detectable blood concentrations of the drug, as has been demonstrated by our group and others (reference in the original version:25). Unfortunately, no serum drug concentrations were analyzed in this study (new comment in the limitations of the study page 7)
As the reviewer comments, we have included the interesting reference for leucine-rich α2 -glycoprotein (LRG)(ref 26 on page 7).
Minor points
1) The IL-6 receptor is rarely described as rIL-6. In a search using PubMed, there are 32 papers that use the term "rIL-6" within the 10 years from 2011 to 2021. With the exception of one episode in which "rIL-6" indicated rat IL-6, the rest were all used in the sense of recombinant IL-6. IL-6 receptor is commonly referred to as IL-6R.
According to the suggestion, we have changed rIL-6 to IL-6R
2) Please check the labels of each table. Some table present "ant-" and the other present "anti-". ("anti- would be correct. Again, the term "anti-rIL-6" is not standard description.)
Thank you, we have made the corresponding correction.

Reviewer 2 Report
The manuscript is interesting and well written. However, I suggest to discuss if hsCRP which is correlated with US synovitis may decreased in patients receiving biologics if those are effectively responding to JAKi and if the patients who will respond may be demonstrated by pharmacogenomics as for other biologics (see and add as references papers by Murdaca et al concerning pharmacogenomics in patients receiving etanercept).
Author Response
The manuscript is interesting and well written. However, I suggest to discuss if hsCRP which is correlated with US synovitis may decreased in patients receiving biologics, if those are effectively responding to JAKi and if the patients who will respond may be demonstrated by pharmacogenomics as for other biologics (see and add as references papers by Murdaca et al concerning pharmacogenomics in patients receiving etanercept).
Thanks for your comment. Our study design (cross-sectional) was not able to evaluate how hsCRP and clinical or US synovitis were changing over time according to treatment (biologics or JAKi). We agree that differences in pharmacogenomics might influence the clinical response. We did not examine differences in hsCRP and clinical or US synovitis according to pharmacogenomics, but this certainly is an interesting area to explore in the future. We have added a brief comment in the limitations of the study about it and add two references including that by Murdaca et al. (Ref 27 and 28 on page 7).

Reviewer 3 Report
This is a clinical study that analyzed whether high-sensitivity CRP (hsCRP) reflected the inflammatory disease status in RA subjects that recived either IL-6 receptor antibodies (anti-rIL-6) or JAK inhibitors (JAKi) treatment, evaluated by clinical and ultrasound parameters.
This is a well-written interesting study, however, a few issues must be addressed.
The introduction should include more details on the chosen therapy to be analyzed, amongst all the possible therapeutic options.
The authors should specify the null hypothesis.
The discussions section is very limited, the authors should include more studies and compare their results to those in the literature.
Author Response
This is a clinical study that analyzed whether high-sensitivity CRP (hsCRP) reflected the inflammatory disease status in RA subjects that recived either IL-6 receptor antibodies (anti-rIL-6) or JAK inhibitors (JAKi) treatment, evaluated by clinical and ultrasound parameters. This is a well-written interesting study, however, a few issues must be addressed the introduction should include more details on the chosen therapy to be analyzed, amongst all the possible therapeutic options.
The chosen therapy (anti- IL-6R or JAKi) was according to the recommendations of the European (EULAR) and National Societies of Rheumatology (SER) for RA management. In general, anti- IL-6R and JAKi are initiated in patients with failure of csDMARDs, specially MTX, and in most cases after failure with a biologic anti-TNF agent. As shown in table 1, more than 50% of patients included in the study were previously treated with other biologics, especially anti-TNF. We have included this statement in the revised version (table 1 page 4).
The authors should specify the null hypothesis.
We thank the reviewer for this observation. We now included our hypothesis in the introduction section page 2. Our null hypothesis was “there is no difference between hsCRP levels and disease activity in RA patients receiving anti-IL-6R therapies or JAKi”.
The discussions section is very limited, the authors should include more studies and compare their results to those in the literature.
There are some studies analyzing the effect of CRP serum levels in assessing disease activity in taking anti-IL-6R that are mentioned in the original report (reference 25)) but no studies previously addressed the exact role of serum CRP in assessing disease activity including ultrasonographic parameters in patients treated with tocilizumab or sarilumab not in patients under JAKi. We have added this statement in the discussion section with new references on the effect of these therapies in ultrasound synovitis of RA patients (new ref 18,19 and 20).

Round 2
Reviewer 1 Report
I checked the new version and confirmed that corrections to the wording and references to new biomarkers have been added. Unfortunately, the data in this paper do not contain any new findings. However, I judge that the data is worth referring by the future generations as negative data.